

# Spatial and temporal trends in dung beetle research

Zac Hemmings[1,2], Maldwyn J. Evans[3] and Nigel R. Andrew[2,4]

[1] Department of Regional NSW, New South Wales Department of Primary Industries, Coffs Harbour, NSW, Australia
[2] Insect Ecology Lab, Zoology, University of New England, Lismore, NSW, Australia
[3] Fenner School of Environment and Society, The Australian National University, Canberra, ACT, Australia
[4] Faculty of Science and Engineering, Southern Cross University, Lismore, NSW, Australia

Corresponding author
Nigel R. Andrew,
nigel.andrew@scu.edu.au

## ABSTRACT

Dung beetles are one of the most charismatic animal taxa. Their familiarity as ecosystem service providers is clear, but they also play a range of roles in a variety of different ecosystems worldwide. Here, we give an overview of the current state of dung beetle research and the changes in the prevalence of topics in a collated *corpus* of 4,145 peer-reviewed articles of dung beetle research, spanning from 1930 until 2024. We used a range of text-analysis tools, including topic modelling, to assess how the peer-reviewed literature on dung beetles has changed over this period. Most of the literature is split into three distinct, but related discourses–the agri/biological topics, the ecological topics, and the taxonomic topics. Publications on the 'effect of veterinary chemicals' and 'nesting behaviour' showed the largest drop over time, whereas articles relating to 'ecosystem function' had a meteoric rise from a low presence before the 2000's to being the most prevelant topic of dung beetle research in the last two decades. Research into dung beetles is global, but is dominated by Europe and North America. However, the research from South America, Africa, and Australia ranges wider in topics. Research in temperate and tropical mixed forests, as well as grasslands, savanna and shrublands dominated the *corpus*, as would be expected from a group of species directly associated with large mammals. Our assessment of dung beetle research comes when ecosystem service provision is becoming more important and more dominant in the literature globally. This review therefore should be of direct interest to dung beetle researchers, as well as researchers working in agricultural, ecological, and taxonomic arenas globally. Research worldwide and across agri/biological, ecological, and taxonomic discourses is imperative for a continued understanding of how dung beetles and their ecosystem services are modified across rapidly changing natural and agricultural landscapes.

## INTRODUCTION

Dung beetles have been a subject of interest to scientists and natural philosophers for centuries. During the 19<sup>th</sup> century, they captured the imagination of famed entomologist Jean-Henri Fabre. In his typical prose, Fabre described the behaviour and life cycle of these

scavenger beetles detailing the lives of the "Sacred beetle", "Spanish Copris", and "the Sisyphus", noting the care with which they provide for their young, a trait most uncommon amongst insects (*Fabre & Henri, 1925*).

> *The peasant of ancient Egypt, as he watered his patch of onions in the spring, would see from time to time a fat black insect pass close by, hurriedly trundling a ball backwards. He would watch the queer rolling thing in amazement, as the peasant of the Provence watches it to this day* (*Fabre, 1918*).

Given this long-held fascination, one may surmise that the dung beetles are a large and prolific group, and highly charismatic (*Ducarme, Luque & Courchamp, 2013*). However, they are a comparatively small group of beetles, with approximately 8,000 described species, comprising ~2% of described beetle species (~387,000 species, *Stork, 2018*) and ~0.8% of described insect species (~1,013,825 species, *Stork, 2018*). The term 'dung beetle' is colloquially used to describe any beetle found inhabiting dung, however, among the scientific community, the term denotes scarab beetles belonging to the family Geotrupidae, and the subfamilies Aphodiinae and Scarabaeinae; with members of the Scarabaeinae often referred to as 'true' dung beetles (*Britannica TEoE, 2024*). The majority of dung beetles rely on the dung of vertebrates, as both a source of food and vital component of their reproductive cycle, however, many also feed on fungi and decomposing materials. There has been significant global interest in dung beetles relative to the size of the group, with thousands of peer-reviewed articles, books (*e.g.*, *Doube & Marshall, 2014*; *Floate, 2023*; *Hanski & Cambefort, 1991*; *Scholtz, Davis & Kryger, 2009*; *Simmons & Ridsdill-Smith, 2011*), and an untold amount of grey literature.

Scientific interest in dung beetles can be attributed to the myriad adaptations that have evolved as a result of the unique ecological niche they inhabit. As obligate coprophages, they are one of the two insect groups, alongside the Dipterans, that feed on and actively break down dung, doing so on a far larger scale than any other group (*Floate, 2023*; *Holter, 2016*; *Losey & Vaughan, 2006*). Processing and relocation of dung facilitate a number of vital ecosystem functions, including nutrient cycling, bioturbation, and seed dispersal (*Nichols et al., 2008*). Dung beetles are a widespread group with representatives inhabiting both xeric and mesic ecosystems, from warm tropical forests and savannahs to hot deserts and temperate rangelands (*Hanski & Cambefort, 1991*). However, it is their potential to complement livestock systems and pastures *via* increases in productivity and decreased management costs that capture the imagination of policymakers (*Beynon, Wainwright & Christie, 2015*; *Herrero & Thornton, 2013*). Livestock grazing systems cover ~26% of the planet's ice-free land area (*Steinfeld, Wassenaar & Jutzi, 2006*) with livestock estimated to consume 4.7 billion tons of biomass per annum, excreting 60–95% of the nutrients present in the original plant matter (*Wilkinson & Lowrey, 1973*). The breakdown of this excrement by dung beetles facilitates the movement of organic matter and nutrients through the soil profile, increasing plant biomass in livestock systems which is subsequently passed on to the livestock themselves (*Doube, 2018*). This process can be controlled and supplemented to improve results or for a more targeted effect, such as the integration of biochar into cattle feed, which is subsequently moved into the soil (*Joseph et al., 2015*).

The reasons that such a small group is so extensively covered across a broad range of literature types can be attributed to not only their scientific and ecological value, but also to their societal value (*van Huis, 2021*). Indeed, their perceived value has grown in recent years as societies are changing perceptions of climate change, and increasing focus on conservation and ecologically sustainable management practices (*Beynon, Wainwright & Christie, 2015*). Agriculture is a major contributor of anthropogenic habit modification and producer of greenhouse gases, accounting for 18% of anthropogenic emissions (*Steinfeld et al., 2006*; *Steinfeld, Wassenaar & Jutzi, 2006*). Post-excretion microbial activity within the dung causes the release of $CO_2$, $NH_3$, $N_2O$, and $CH_4$ (*Clemens & Ahlgrimm, 2001*), resulting in soil acidification, eutrophication, ozone depletion, and strengthening the green-house effect of the atmosphere, resulting in an increase in mean global temperature and perturbed weather patterns. There has been increasing evidence, that by breaking up dung and disrupting the anaerobic conditions required by gas-producing microbes, dung beetles alter the profile of greenhouse gasses released into the atmosphere (*Iwasa, Moki & Takahashi, 2015*; *Penttilä et al., 2013*; *Piccini et al., 2017*; *Slade et al., 2016*). Recent evidence has also suggested that activity by tunnelling dung beetle species reduces the impact of drought conditions on plant growth by increasing soil water retention (*Johnson et al., 2016*). This reduction in drought conditions may provide a potential avenue for biological mitigation of the effects of climate change (*Johnson et al., 2016*; *Maldaner et al., 2021*).

The planet is currently experiencing previously unseen anthropogenic disturbance and ecosystem modification (*IPBES, 2019*; *Newbold et al., 2015*; *Western, 2001*). The use of bio-indicator taxa to monitor the health of ecosystems has gained prominence among ecologists and conservationists (*Evans et al., 2019*; *Holt & Miller, 2010*). Dung beetles have proven to be an ideal bio-indicator due to their well-described diversity, strong links to environmental processes, global distribution, and reliance on other organisms in the community (*Raine & Slade, 2019*). For these reasons, dung beetles provide the ability to monitor changes in ecosystem function over time easily and at little financial cost (*Spector, 2006*).

The aim of this review is to provide an overview of the temporal and spatial trends of dung beetle research. This review will be of direct interest to dung beetle researchers, as well as researchers working in agricultural, ecological, and taxonomic arenas globally. This manuscript can assist in identifying knowledge gaps to help dung beetle researchers identify areas that need further study, ensuring that future research is directed where it's most needed. Further, it will assist agricultural and ecological researchers. Understanding how dung beetles contribute to ecosystem services like nutrient cycling and soil aeration can inform sustainable land management practices. This manuscript can also assist researchers track how dung beetle research behaviors are changing over time and space, which is crucial for assessing the impacts of climate change, land-use changes, and species introductions. Additionally, it will help taxonomic researchers benefit from understanding the broader ecological and agricultural contexts, which can lead to more comprehensive and applicable taxonomic studies. Finally, given the global nature of these challenges, such a review provides a valuable comparative perspective that can be applied in different regions and contexts.

**Peer**J

Our intention is not to provide a comprehensive analysis of evidence from the literature, as there are a number of books that summarise the current state of knowledge and provide excellent syntheses (*Hanski & Cambefort, 1991*; *Scholtz, Davis & Kryger, 2009*; *Simmons & Ridsdill-Smith, 2011*). Rather, we used a combination of text-anaysis techniques to elucidate temporal and spatial trends in dung beetle research (*Andrew & Evans, 2023*; *Andrew et al., 2022*; *Evans et al., 2021*, *2023a*, *2022*, *2023b*). Firstly, we used structural topic modelling to reveal the dominant topics in the *corpus* (*Roberts et al., 2014*). We then carried out several *post-hoc* analyses to explore the trajectories and similarities of these topics (*Westgate et al., 2015*). Following this, we then combined our topic model with geoparsing and taxonomic entity extraction (*Andrew & Evans, 2023*; *Evans et al., 2023a*; *Millard, Freeman & Newbold, 2020*).

Specifically, we addressed the following questions:

1) What are the key research topics that have been carried out using dung beetles?
2) How has topic prevalence changed over time?
3) How are these topics distributed globally?
4) How are key topics aligned with differed biomes globally?
5) What are the key taxa associated with dung beetle research?

## METHODS

Portions of this text are accessible pre-publication as part of a preprint (*Hemmings, Evans & Andrew, 2024b*) and a PhD Thesis (*Hemmings, 2018*).

### Literature search

We chose to limit our search to peer-reviewed articles. We acknowledge that there is a large amount of grey literature and other publications, such as books on the subject of dung beetles. However, much of this literature is not accessible to a global audience and is not peer reviewed. As a result, we targeted the most scientifically robust and readily available publications, an approach which fits well within our aim to explore the temporal and spatial trends of dung beetle research.

We searched the Scopus and Web of Science literature indexers using the following Boolean search terms: dung & beetle*; scarabaeinae; aphodiidinae; geotrupinae; coprophag* & beetle*; coprophag* & scarab*; coprophag* & coleop*. These results were combined with another Endnote library compiled by the authors, which consisted of articles published from 1933 to 2017. This library consisted of articles retrieved from Scopus and Google Scholar using the search terms: dung beetle; scarabaeinae; aphodiinae; geotrupinae. Web of Science returned a total of 3,179 publications, Scopus returned a total of 3,140 publications, and the existing library contained 1,448 publications. After removing duplicates, the final *corpus* contained 4,145 articles published between 1933 and January 2024. The data file used here can be found on Figshare (*Hemmings, Evans & Andrew, 2024a*).

## Topic modelling

To prepare the *corpus* for topic modelling, we used the 'textProcessor' function in the 'stm' package (*Roberts, Stewart & Tingley, 2019*) in R (*R_Core_Team, 2023*), to remove punctuation, stop words, numbers, and words with fewer than three characters. We also stemmed words to their root form (*e.g.*, walk = walked, walking, walker) and removed the most rare (in <1%) and common (in >85%) words in each abstract and title.

We then fitted a structural topic model (STM) using the 'stm' package, analysing abstracts and titles of our *corpus*. After trialling a number of topics, we chose 20 topics as a number large enough to provide sufficient detail to analyse the topic landscape of our *corpus*, but not too large as to be overly complex (*Andrew et al., 2022*; *Evans et al., 2023b*; *Westgate et al., 2015*). We used spectral initialisation for model fitting (*Roberts, Stewart & Tingley, 2016*). We then gave our topics a short summary title by examining the 20 highest-weighted words of each topic (*Westgate et al., 2015*).

## *Post-hoc* topic analyses

To examine the temporal prevalence of topics over time, we used the 'estimateEffect' function in the 'stm' package, to fit topic prevalence through time, treating year as a linear term. We used the 'hot' and 'cold' topic nomenclature to describe topics as either increasing or decreasing in prevalence over time, respectively (*Evans et al., 2021*; *Westgate et al., 2015*). We examined topic similarity by undertaking a hierarchical cluster analysis using Ward's minimum variance method on the dissimilarities of the model-derived probabilities of word occurrence matrix (*Ward, 1963*). We then grouped these topics into six groups of closely related articles based on this clustering (*Evans et al., 2023b*). We also explored how each topic was distributed through the *corpus* by examining the specificities and generalities of each topic within the whole *corpus* (*Westgate et al., 2015*). Topics that are considered general would have topic weights that span multiple topics, whereas topics that are considered specific would have topic weights heavily biased towards one topic. To calculate this, we assigned each article to its highest-weighted topic and calculated mean weights for each topic of those articles selected, *vs* those that were not selected (*Westgate et al., 2015*).

## Taxanomic entity extraction

We used the Global Names Finder v1.1.3 (https://finder.globalnames.org/) to scan for taxonomic mentions in all abstracts and titles in the *corpus*. We then fetched genera, orders, classes, and kingdoms of all taxonomic names, using the National Center for Biotechnology Information database Application Programming Interface (API) through the 'taxize' package (*Chamberlain & Szöcs, 2013*) in R.

## Geoparsing

We scanned all article abstracts and titles for geographic mentions using the CLIFF-CLAVIN geoparser (*D'Ignazio et al., 2014*; *Millard, Freeman & Newbold, 2020*) in Python using a Docker container (*Merkel, 2014*) hosted on the GitHub repository (<https://github.com/havlicek/CLIFF-docker>). CLIFF-CLAVIN is able to resolve

mentions to the most likely physical coordinates. We categorized the mentions into 'minor' mentions (specific locations within countries) and 'major' (countries) (*Millard, Freeman & Newbold, 2020*). We then assigned World Wide Fund for Nature (WWF) biomes for all the minor mention locations (*Olson & Dinerstein, 2002*).

## RESULTS AND DISCUSSION

### Q1. What are the key research topics that have been carried out using dung beetles?

We based our assessment on 20 topics (Table 1) across the entire dung beetle *corpus* based on the word clouds generated (Appendix 1). The 20 topics split into six broad dendrogram clusters (Fig. 1): three large clusters and three single-topic clusters. The largest topic cluster (Cluster #1) was agricultural and biologically focused. It included 'Nesting behaviour', 'Food preference and diet', 'Navigation', 'Statistical analysis', 'Biomimetics', 'Soil health and plant growth, 'Agricultural associations', 'Conservation & biodiversity', and 'Abiotic response variables'. This is a broad and diverse cluster primarily driven by a high overlap of keyword usage. Four subgroups with more ecological alignment come out within the group: Subgroup 1: 'Nesting behaviour', and 'Food preference and diet' relate to biological interactions the beetles have with dung, and their preference for offspring food provision. Subgroup 2: 'Navigation', 'Statistical analysis' and 'Biomimetics' are aligned with data analysis and statistics (*da Silva, Mota Souza & Neves, 2022*), methods of assessing dung beetle movement (*Dacke et al., 2013*), and the use of dung beetles in the development of new technologies (*e.g., Tong et al., 2005*; *Wang et al., 2018*). Subgroup 3 'Soil health and plant growth' and 'Agricultural associations' directly aligns with the roles that dung beetles play in farm productivity (*Beynon, Wainwright & Christie, 2015*; *Doube, 2018*). Subgroup 4: 'Conservation and Biodiversity' and 'Abiotic response variables' aligns with research in native environments and ecological research relating to aspects such as climate change (*Maldaner et al., 2021*).

The second largest topic cluster (Cluster #6) was ecologically focused. It included 'Ecosystem function', 'Landscape ecology', 'Species distributions', 'Sampling', and 'Assemblage structure'. This was a clear cluster around the ecological functions dung beetles provide that are key to the continued existence of many habitats (*Noriega et al., 2023*). The provision of these functions also provides a number of services that directly benefit humans (*deCastro-Arrazola et al., 2023*; *Nichols et al., 2008*), as well as the way the dung beetles are collected (*Heddle, Hemmings & Andrew, 2023*) and the associated assemblage structures (*Noriega et al., 2021*).

The third largest topic cluster (Cluster #2) was taxonomically focused. It included 'Taxonomy', 'Phylogeny', and 'Scarabaeidae' clustered together. This was a clear taxonomic grouping based on species naming and descriptions. One of the key elements to this grouping was the taxonomic name change of *Onthophagus* to *Digitonthophagus* in 2017 (*Génier & Moretto, 2017*). Additionally, species descriptions and taxonomic changes have been especially active (*Cupello, Silva & Vaz-de-Mello, 2023*).

**Table 1 Twenty uncovered topics from dung beetle research articles.**

| Topic no. | Title | Description and key words |
|---|---|---|
| 1 | Nesting behaviour | Aligned with brood, larvae, egg, ball, nest. |
| 2 | Phylogeny | Aligned with phylogenetic*, endem*, morphology, Africa*, and sequenc* |
| 3 | Seed dispersal | This topic is clearly referring to dung beetle mediated secondary seed dispersal. The topic is most strongly associated with the word seed, followed by dispers*. |
| 4 | Soil health and plant growth | Clear alignment with the words soil, plant, decomposition and nutrient. |
| 5 | Sexual selection and sexual traits | Clear alignment with the words male, femal* horn, reproduct* and size |
| 6 | Effect of veterinary chemicals | Aligned with ivermectin, treatment, product and residu* |
| 7 | Ecosystem function | Clear alignment with the words ecosystem, function, community* and divers* |
| 8 | Scarabaeidae | Alignment with scarabaeida*, onthophagus and speci* |
| 9 | Conservation & biodiversity | Alignment with the words biodiverse* and conserve*. |
| 10 | Taxonomy | New species descriptions and taxonomic determinations. Strongly associated with the words speci*, new and scarabaeida. |
| 11 | Landscape ecology | Alignment with the words forest, landscap*, habitat, and fragment. |
| 12 | Food preference and diet | Alignments with the words food, feed, attract, and resource*. |
| 13 | Species distributions | Aligned with distribution*, assemblage*, divers*, gradient |
| 14 | Sampling | Aligned with season, trap, sampl* |
| 15 | Statistical analysis | Aligned with model, method, data |
| 16 | Abiotic response variables | Aligned with chang*, respons*, temperatur* |
| 17 | Agricultural associations | Aligned with cow, sheep, pastur* and import |
| 18 | Assemblage structure | Aligned with community* and habitat |
| 19 | Navigation | Aligned with orient*, direct*, flight, and light |
| 20 | Biomimetics | Aligned with water, structure, properti* |

**Note:**
* indicates indicates search wildcard where we searched for multiple forms of the word ending.

The topics 'Seed dispersal', 'Sexual selection & sexual traits', and 'Effect of veterinary chemicals' formed unique topic clusters (respectively Cluster #3, #4, and #5). All three are unique and distinctive topics. 'Seed dispersal' is an explicit ecosystem service that dung beetles provide (*Manns et al., 2020*; *Midgley et al., 2015*). 'Sexual selection and sexual traits' has focused on male selection for reproductive success at both the adult (*Kotiaho et al., 2003*) and sperm level (*House & Simmons, 2006*). 'Effect of veterinary chemicals' is aligned with the loss of ecosystem service provisions provided by dung beetles in agricultural ecosystems based on the use of antiparasitic drugs on cattle, sheep and other livestock (*Mackenzie et al., 2021*; *Verdú et al., 2020*; *Weaving, Sands & Wall, 2020*).

### General vs specific topics

General topics (Fig. 2) are found in the bottom right-hand corner and specific topics are found in the top left corner. 'Agricultural associations' and 'Assemblage structure' were defined as a general topics. Specific topics included 'Seed dispersal', 'Biomimetics',
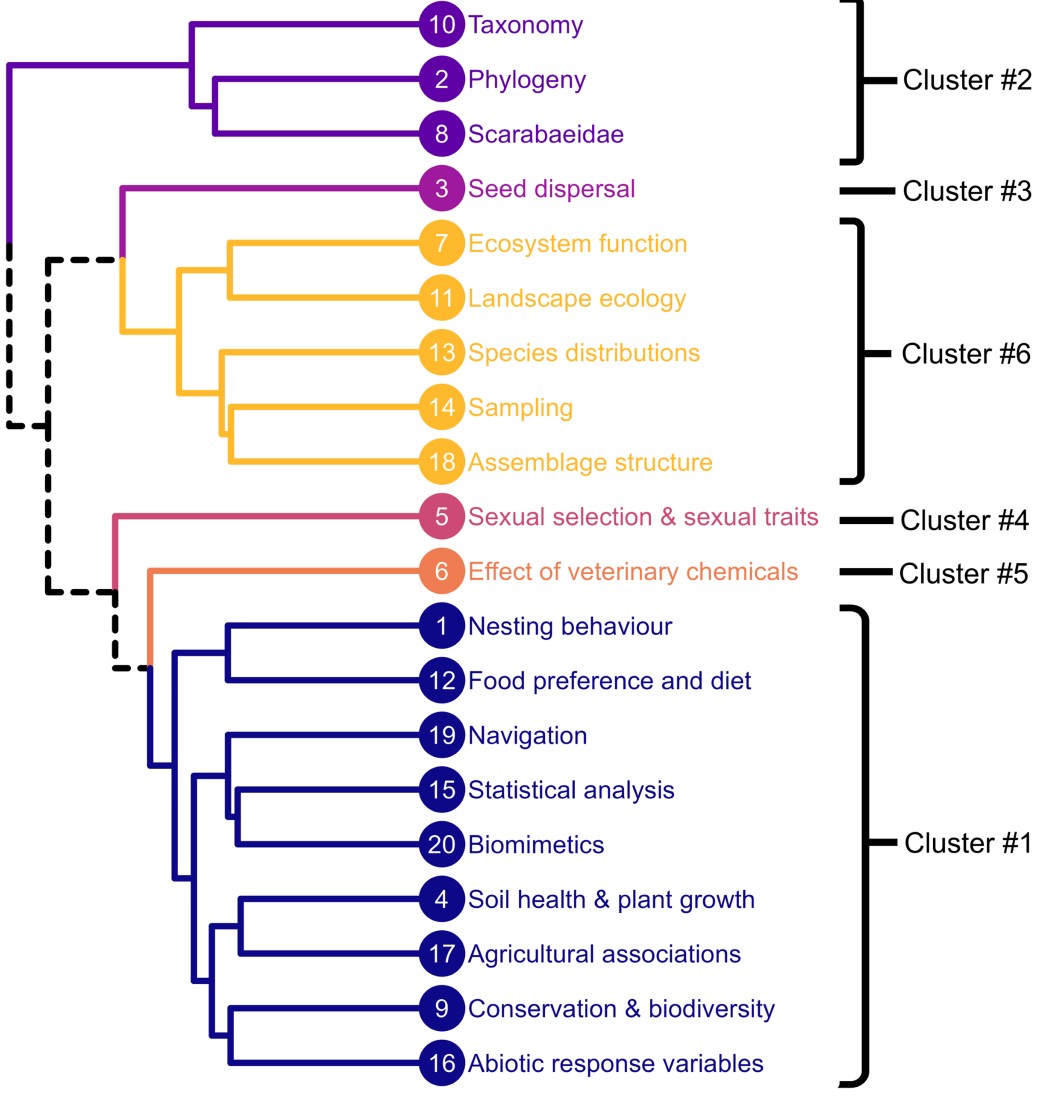

**Figure 1 Dendrogram showing relationships between the 20 Dung beetle topics generated.** The six clusters are represented by different colours. Solid line represent specific clusters. Dotted line represents where branches between clusters meet.

'Sexual selection & sexual traits' and 'Taxonomy'; 'Effect of veterinary chemicals' was weakly defined as a specific topic.

## Q2. How has topic prevalence changed over time?

Articles published on dung beetles have exhibited an exponential increase over time (Fig. 3A), with a rapid rise after the year 2000: this trend is consistent with other insect-related publications (*e.g.*, *Andrew et al., 2013*). Topics showed a range of temporal trends over the period of research assessed (1933 to 2024) (Fig. 4). 'Ecosystem function' had the largest increase in prevalence over the time period. Modest increases in prevalence were found in topics including 'Landscape ecology', Conservation & biodiverstiy', 'Abiotic response variables', 'Statistical analysis', and 'Species distributions'. No change in

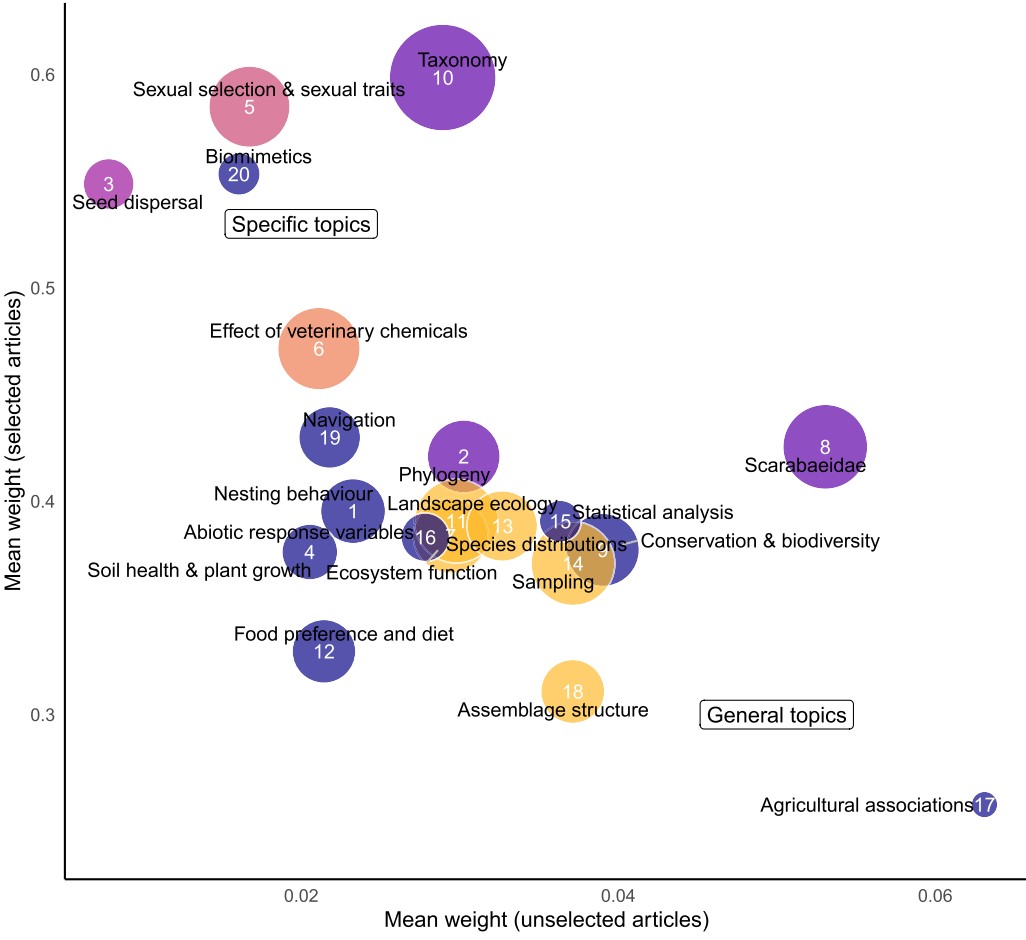

**Figure 2  Representation of how each topic is distributed within the *corpus*.** Topics in the bottom right corner are comparatively distributed evenly through the *corpus*, whereas topics in the top left corner are heavily weighted towards one topic.                

prevalence was found in 'Taxonomy', 'Phylogeny', 'Sampling', 'Biomimetics', 'Soil health & plant growth', 'Seed dispersal', 'Food preference and diet', and 'Sexual selection & sexual traits'. Topics that reduced in prevalence include 'Agricultural associations', 'Assemblage structure', Scarabaeidae' and 'Navigation'. The largest reduction in prevalence during this period included 'Nesting behaviour' and 'Effect of veterinary chemicals'. 'Nesting behaviour' had been a popular topic (top 4) up until the 1990's and 'Veterinary chemicals' had been the most popular topic in the 1970s–1990's (Fig. 3). This was likely caused by i) research for the Australian dung beetle project (*e.g., Bornemissza, 1976*), and ii) the necessity for fundamental research into the behaviour and biology of common species (*e.g., Ridsdill-Smith, 1988, 1993; Ridsdill-Smith, Hall & Craig, 1982*), which once undertaken, have paved the way for more complex research on the interactions between functional groups and their effect on the environment. 'Ecosystem function' only emerged as a topic in the 2000's and had a relatively meteoric rise over the following two decades (Fig. 5). While ecosystem services were studied prior to 2000 (*Bornemissza, 1976; Hughes,*
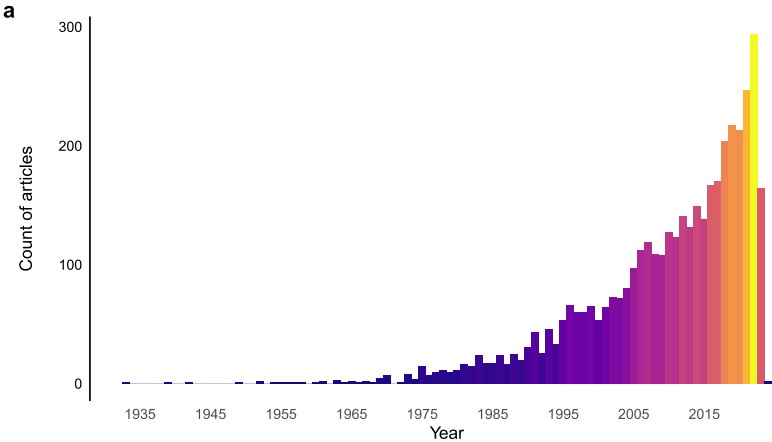

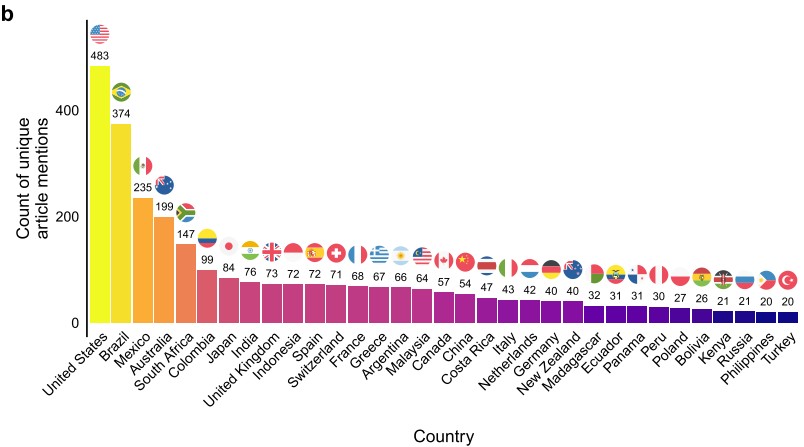

**Figure 3** (A) Number of dung beetles articles published per year and (B) number of article published overall from each location mentioned in the manuscript.

*1975*), it was not until post 2000 that the term 'ecosystem function' saw common usage in the literature, but the terminology used was different.

## Q3. How are these topics distributed globally?

Most of the topics were dominated by research emerging from North America and Europe (Figs. 3B, 6). This is an example of the Global North domination in research published in English across a range of biological disciplines (*Ballari et al., 2020*; *Piguet, Kaenzig & Guélat, 2018*), but more broadly across a range of research diciplines (*Collyer, 2018*; *Oztig, 2022*), as well as issues of publishing using English language (*Haelewaters, Hofmann & Romero-Olivares, 2021*; *Zenni & Andrew, 2023*). The most 'global' topics include 'Nesting behaviour', 'Taxonomy', 'Landscape Ecology', 'Species Distributions', and 'Sampling'. All topics are represented in North America, Western Europe and Africa. Research topics not covered in Eastern Europe include 'Phylogeny', 'Seed dispersal' 'Sexual selection & sexual traits', 'Effect of veterinary chemicals', 'Ecosystem function', 'Conservation & biodiversity', 'Species distributions', 'Statistical analysis', 'Agricultural associations', 'Assemblage structure', and 'Biomimetics'. Topics not covered in Asia, excluding Japan is 'Sexual selection and sexual traits'; and including Japan is 'Statistical analysis'. Not included in

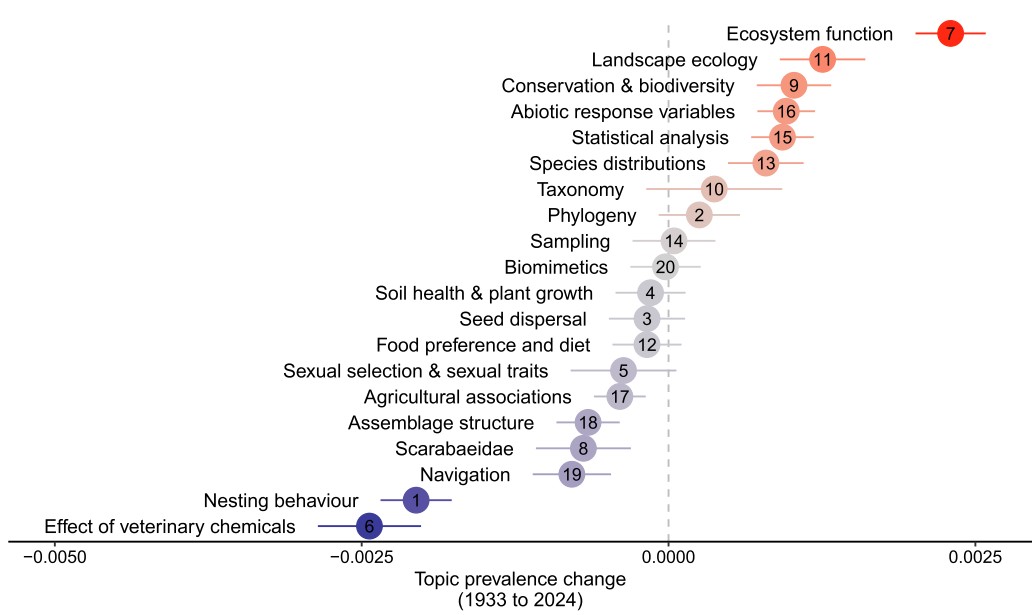

**Figure 4** Change in the prevalence of topics in the dung beetle literature *corpus* from 1930–2021.

**Figure 5** Bumpplot showing the relative change in the publication of topics across each decade from 1930 to 2020.

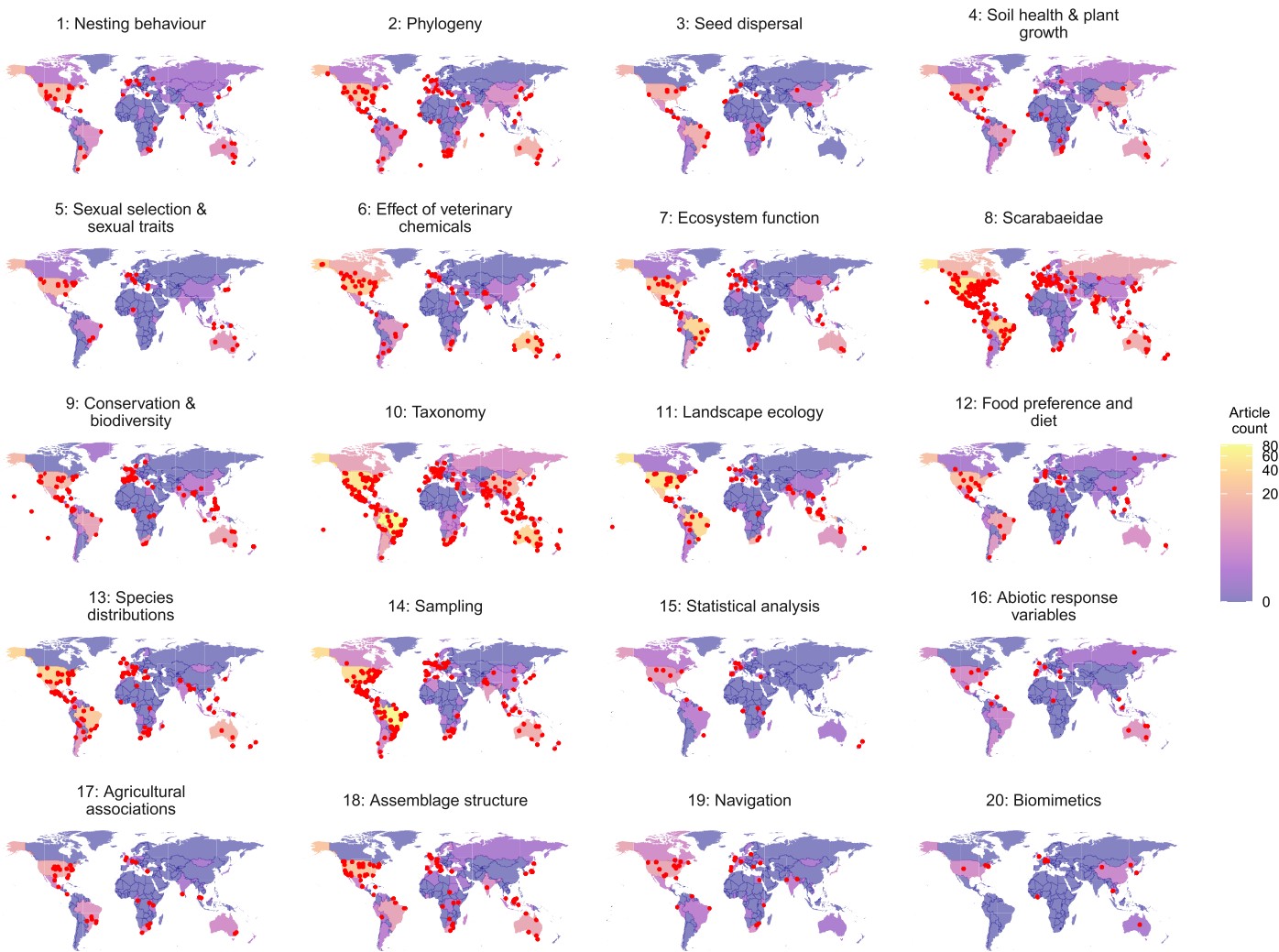

**Figure 6** **Location of dung beetle publications across each of the 20 topics.** Country colour indicative of topic prevalence.

Australasia include 'Seed dispersal' and 'Navigation'. 'Biomimetics'—the study of nature that may benefit science, engineering, and medicine more broadly (*Fayemi et al., 2017*)—was not a topic covered in South America.

## Q4. How are key topics aligned with different biomes globally?

'Scarabaedae' and 'Taxonomy' topics had a strong association with the Temperate Broadleaf & Mixed Forests, and an association with Tropical & Subtropical Moist Broadleaf Forests and Deserts & Xeric Shrublands WWF biomes (Fig. 7). 'Landscape ecology' was strongly associated with Tropical & Subtropical Moist Broadleaf Forests. Mangroves, Boreal Forests/Taiga, Tundra and Flooded Grasslands & Savannas were not represented. The other topics had few, if any, mentions/studies with the remaining WWF biomes.

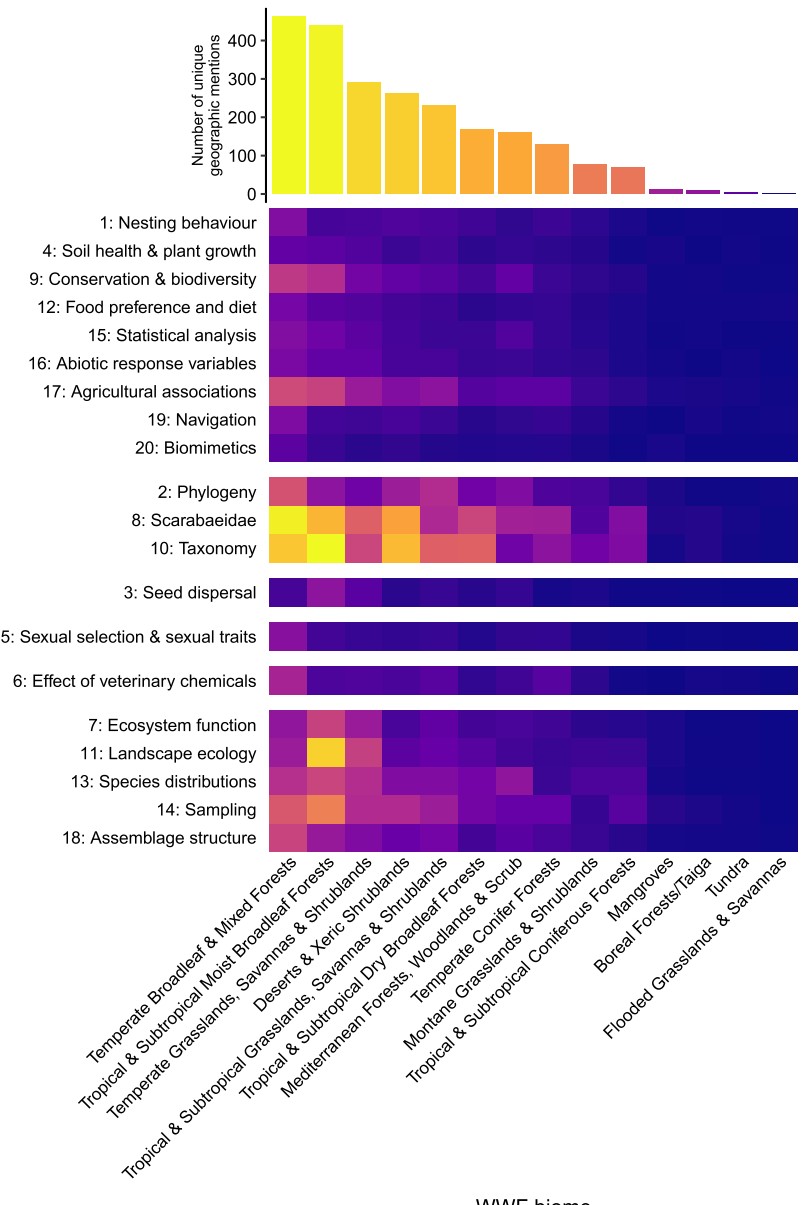

**Figure 7 Biome plot indicating the number of mentions of each biome across the 20 topics.** Topics grouped into their six clusters (as per Fig. 1). Cell colour indicative of topic prevalence.

## Q5. What are the key taxa associated with dung beetle research?

*Onthophagus* was the most studied genera (483 mentions) in the articles that we analysed (Fig. 8A). This reflects *Onthophagus'* diversity and global distribution: it is among the most speciose genera in the animal kingdom, with ca. 2,300 species (*Breeschoten et al., 2016*). Additionally, *Onthophagus* is a model organism for the study of the evolution of sexual dimorphism and the development of male horns (*Kijimoto et al., 2013*; *Moczek, 2011*). *Aphodius, Copris,* and *Canthon* were a distinct cluster of second-ranked studies (157, 147 and 146 mentions respectively). *Dichotomius, Euoniticellus, Scarabaeus, Digitonthophagus,*

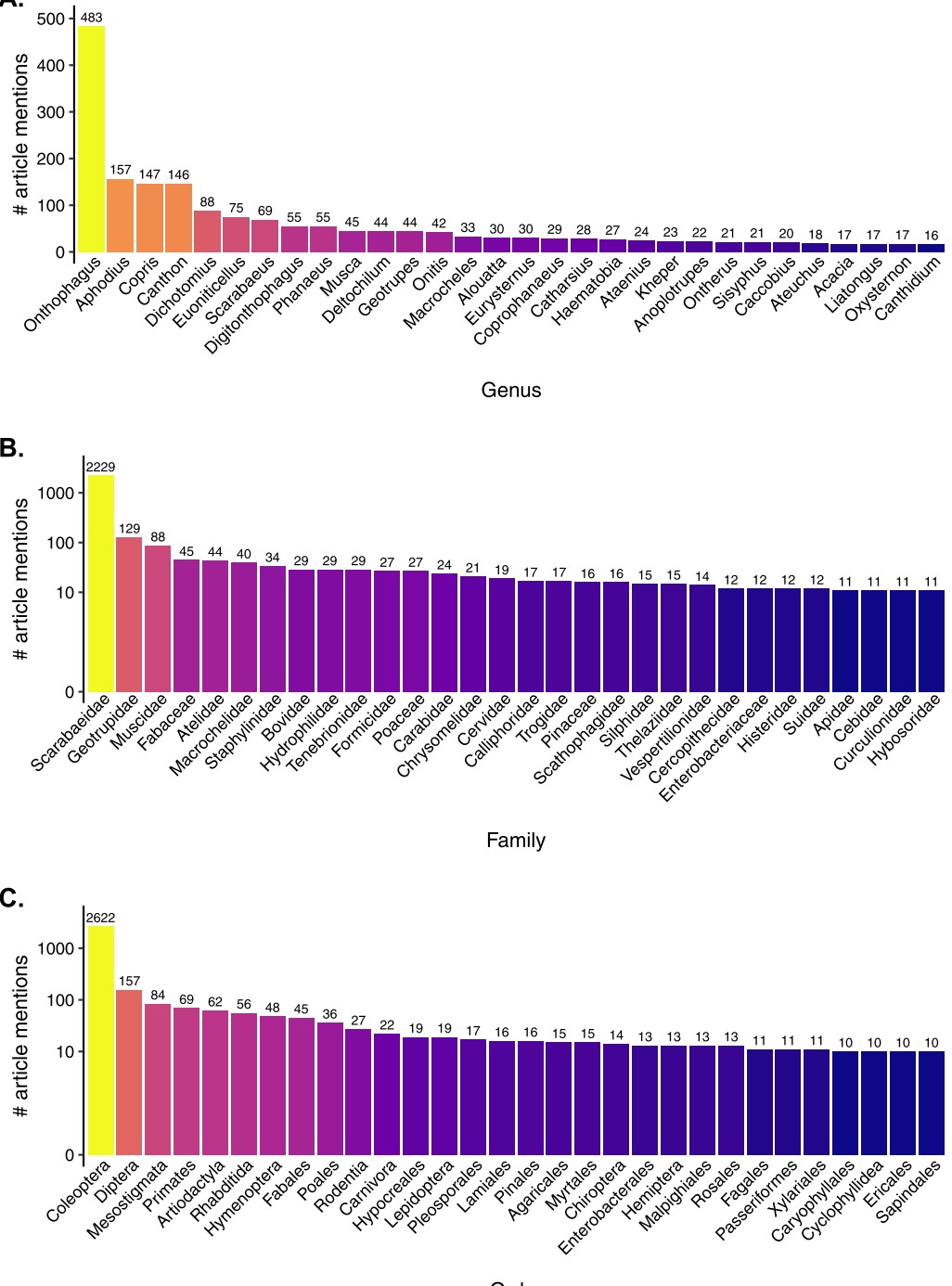

**Figure 8** **Taxonomic mentions across all topics: (A) Genus level, (B) Family level, (C) Order level.** Bar colour indicative of topic prevalence.

and *Phanaeus* had between 88 and 55 mentions. As mentioned previously, taxonomic reclassifications will influence the assessment of species and genera names (*Cupello, Silva & Vaz-de-Mello, 2023*; *Génier & Moretto, 2017*).
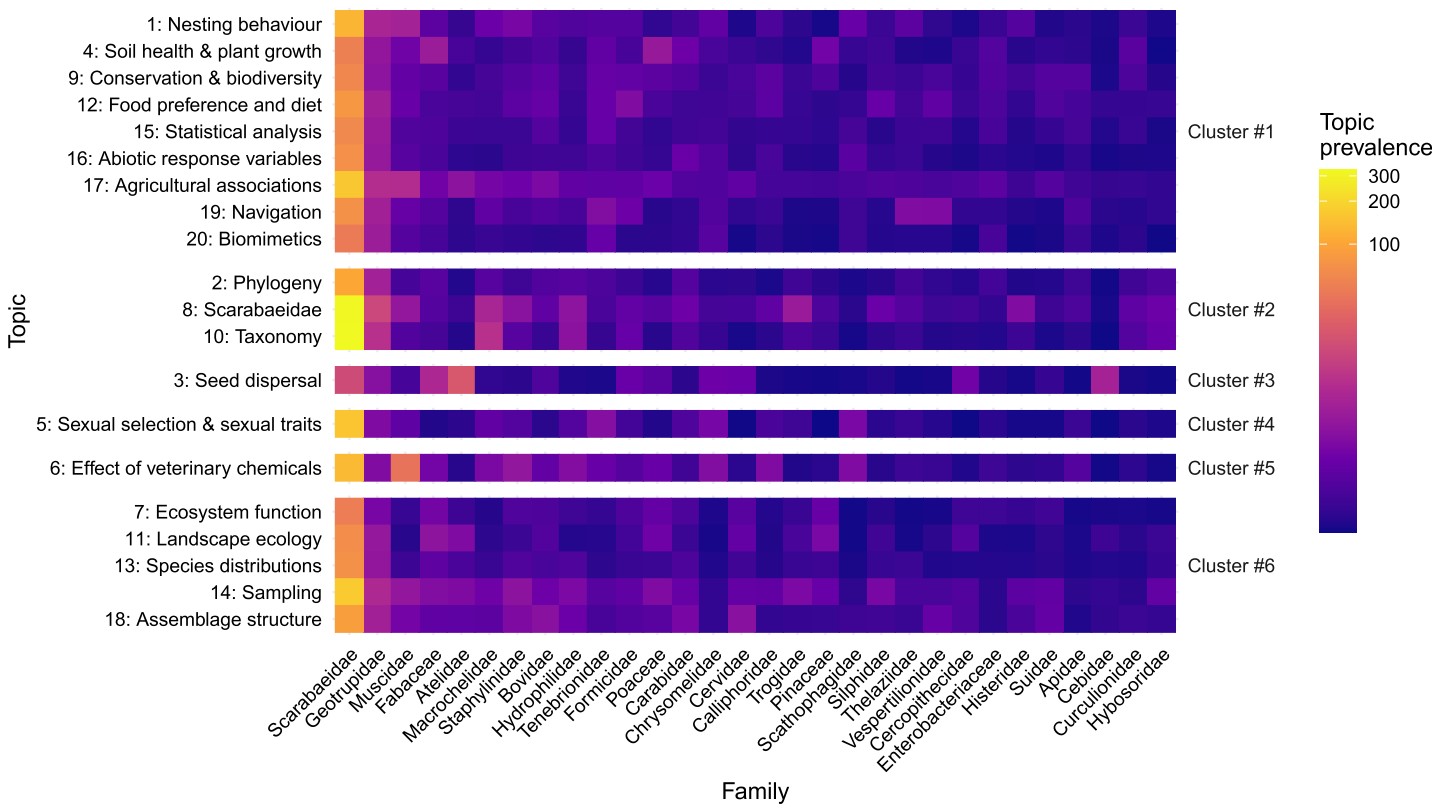

**Figure 9 Heatmap of family mentions across the 20 topics.** Topics groups into Clusters as Per Fig. 1. Cell colour indicative of topic prevalence.

The two dominant dung beetle families 'Scarabaeidae and Geotrupidae' had the most mentions (2,229 and 129 respectively, Fig. 8B). Muscidae were the next common group (88 mentions). At the order level, Coleoptera were the most dominant (2,622 mentions; Fig. 8C), followed by Diptera (157 mentions), Mesostigmata, and Primates (84 and 69 respectively). Due to the strong commensal relationship between dung beetles and vertebrates, there is often extensive co-evolution between the two groups, with dung beetle communities inextricably linked to the historic and contemporary structure of the associated vertebrate community (*Bogoni et al., 2016*). The availability of multiple types of dung in an ecosystem is strongly correlated with the tribal diversity of the associated dung beetle community, with abiotic processes having a stronger influence on diversity at the generic and species levels (*Davis & Scholtz, 2001*). The decline of large-bodied dung beetles in Europe has been proximally tied to the extinction of local megafauna and the subsequent loss of large, wet dung as resource (*Schweiger & Svenning, 2018*), with a similar situation thought to have occurred in Australia (*Doube, 2018*). Changes in the composition of the vertebrate community can have strong effects on dung beetles of a particular functional group. The density of deer was associated with an increase in the abundance of small bodied dung beetle species (<10 mm), whereas the abundance of large bodied species (>10 mm) was unaffected (*Iida, Soga & Koike, 2018*). In Panama areas of forest, dung beetle communities differed between areas with no hunting and areas where monkeys were

hunted; hunted fragments showing decreased species diversity, with the abundance of nocturnal beetles negatively correlated with the abundance of mammals (*Andresen & Laurance, 2007*). Flow-on effects from this can further influence the community, as the presence/proportion of larger, more dominant species of dung beetle influences the structure of smaller species *via* size-asymmetric competition (*Horgan & Fuentes, 2005*).

With family associations with topics, Scarabaeidae had a high or medium prevalence with all topics, (Fig. 9), Geotrupidae had a weak association with all topics, and Muscidae had a medium association with 'Effect of veterinary chemicals'. Dung beetles are known to control muscid fly abundance in agricultural dung (*Kirk, 1992*; *Smith & Matthiessen, 1984*), and the influence of agro-chemicals can reduce the survival, growth and development of dung beetles (*Mackenzie et al., 2021*). All other families had a weak or no association with the 20 topics identified.

## CONCLUSIONS

Dung beetle research is a rapidly developing field with a long and fruitful history. It encompasses taxonomy, fundamental biology, and applied research globally. In this review, we targeted primarily literature found in peer-review web searches–so we do acknowledge that there is also an absence of non-peer-reviewed literature and grey literature that could influence our findings (*Haddaway et al., 2020*; *Paez, 2017*), particularly older material (*Hong et al., 2022*; *Pollman, 2000*). However, it is clear that in the past two decades, 'ecosystem functioning' has become a key area of interest. Dung beetle research will continue to grow–they are key ecosystem service providers globally (*deCastro-Arrazola et al., 2023*; *Noriega et al., 2023*). They serve as an ideal model for addressing ecological, evolutionary, and agricultural questions. Global research, particularly in the Global South and Asia, across agricultural, biological, ecological, and taxonomic discourses, is crucial for understanding how dung beetles and their ecosystem services are affected by land-use change, climate change, and new species introductions.

## ACKNOWLEDGEMENTS

Grammarly, CoPilot and Consensus were used to draft and edit this article.

### Funding

The authors received no funding for this work.

### Competing Interests

Nigel Andrew is an Academic Editor for PeerJ.

### Author Contributions

- Zac Hemmings conceived and designed the experiments, performed the experiments, authored or reviewed drafts of the article, and approved the final draft.
- Maldwyn J. Evans analyzed the data, prepared figures and/or tables, authored or reviewed drafts of the article, and approved the final draft.

- Nigel R. Andrew conceived and designed the experiments, authored or reviewed drafts of the article, and approved the final draft.

## Data Availability

The data is available at figshare: Hemmings, Zac; Evans, Maldwyn; Andrew, Nigel (2024). Spatial and Temporal Trends in Dung Beetle Research dataset. figshare. Dataset. https://doi.org/10.6084/m9.figshare.26103964.v2.

## Supplemental Information

Supplemental information for this article can be found online at http://dx.doi.org/10.7717/peerj.18907#supplemental-information.

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
