# Peer review of "Spatial and temporal trends in dung beetle research"

_PeerJ, doi:10.7717/peerj.18907_

## Round 0.1 · original submission · Major Revisions

Dear authors, please improve this manuscript according to the reviewers' comments. This will improve the perception of your manuscript by readers, make it more understandable. I ask you not to ignore any of the reviewers' comments, all shortcomings must be eliminated.

·

Basic reporting

Coprophagous beetles play an important role in maintaining the ecological balance of ecosystems. These insects are represented by complexes of taxonomic groups. Their existence is inextricably linked with the processing of organic waste of animal origin. Dung beetles have a number of behavioral adaptations that allow them to survive in a competitive environment. Therefore, publications summarizing scientific literature on the study of dung beetles are welcome. The topic of the article under review is very relevant. The research questions in the manuscript are clearly defined. The results and conclusions are supported by statistical analysis. However, there are minor shortcomings and technical comments.

Abstract:
Line 34. I recommend rephrase the sentence. Charisma is a set of emotional and mental abilities of a person. I recommend replacing the word "charismatic".
Lines 55-62. It is better to place keywords in a line.
Introduction:
The authors reveal the essence and problems of the topic. Some parts of the introduction are written well and concisely. Some sentences are too long, difficult for the reader to understand. I recommend breaking it into parts (lines 114-116, 120-123, and others).
Pay attention to punctuation marks. In quotations, a comma must be placed after the author's last name before the year. This error is observed throughout the text.
Lines 76-81, 102-105 statements must be supported by literary citations.
What were your principles when choosing the research topic, please justify? Why are beetles coprophagous, and not Coleoptera, which belong to other trophic groups?

Experimental design

Materials and methods:
Why did the authors use a time period for analyzing scientific literature starting from 1933, and not from 1924?
Line 158. Quotes in brackets are incorrectly formatted. If a publication has more than two authors, et al., year of publication are placed after the first author's last name).
Lines 158, 162. Repetition of the same quote. Combine the text and leave only one reference.
Results and discussion:
The results and discussion are well-reasoned, demonstrating the scientific merit of the manuscript. The data analysis is appropriate, the results are reliable.
Line 320. I recommend rephrasing "nuisance muscid flies".

Validity of the findings

Conclusions:
Lines 326-333. I recommend breaking the sentences into several parts (too long, difficult for the reader to understand).

Additional comments

No comments.

Reviewer 2 ·

Basic reporting

1) The English text is correct, thence there are not major issue to be emended. No changes are required.
2) The online references are more than enough, and the ones quoted in the Reference section of the paper are correct. There is a major issue since authors declared that they examined only peer-reviewed literature. It could be an issue for the older references. Furthermore, how did they assess whether the older papers were peer-reviewed? Examining the figures, it seems that a large part of information is missing for the years 1930-50 at least. Likely, a large part of older taxonomic books were not peer-reviewed, thence a large part of the reference data are missing. How many books published between 1930 and 2000 were included in the analysis (if any was included at all)?
3) The structure of the manuscript is acceptable, although the aim of the research is not entirely clear. Tables and figures are of good quality. Raw data (list of the works examined) can be accessed online.
4) Authors stated that the review will be of direct interest to dung beetles researchers, but no details are given. As as historical review about the changes of dung beetles researches along a century could be interesting, but a more thorough analysis could be added

Experimental design

1) The manuscript is a review of the different research fields about dung beetles, being a Literature Review it is within the aim and scopes of this journal
2) High technician and ethical standard do not apply to the present research
3) Method are described in details and can be replicated
4) some of the older literature could be missing (see above) Authors should give a detailed reply, and include more information in the Introduction
5) Only part of the papers examined are quoted in the References, but the list of the sources examined can be accessed online
6) the review is correctly organized, no changes are required

Validity of the findings

1) The impact of the manuscript is average. The review could be of interest for dung beetles researches, but it will likely not influence the decision to favor one research topic over another. It will be useful to have a list of the last century literature, although part of the older references could be missing.
2) Conclusions section is linked to the original research design, no unresolved questions were identified.

Cite this review as

·

Basic reporting

The English language is professional and unambiguous, although some of the wording is unclear and repetitive, particularly the conclusion e.g. lines 326-329 could be reworded to make it clearer and more concise. Line 242 is not immediately clear you are talking about Figure 2 - it would be clearer to introduce the figure first in some way before launching into a description of the figure itself. Check figure legends for clarity e.g Figure 3 should be 'number of dung beetle papers' and 'number of papers published'.

Introduction and background are good and show to context of the paper. Some key literature may be missing e.g. the recent book by Floate (Floate, KD. 2023. Cow patty critters: An introduction to the ecology, biology and identification of insects in cattle
dung on Canadian pastures. Agriculture and Agri-Food Canada, Lethbridge, Alberta, Canada. 224 pp).

Article structure is good, the review is of broad interest and within the scope of the journal. This review takes a different point of view than other reviews in the field because it studies the spatial and temporal patterns of dung beetle research and identifies key themes over time and geographical space. Introduction is good.

Experimental design

Article is within aims and scope of journal, the methodology is rigorous, sound, and presentation of the results and figures is clear, except figure 6 was difficult to interpret because the maps were so small. Although this is not a systematic review per se, there is a comprehensive coverage of the subject and the methods align with the aims of the paper which are to give an overview of the spatial and temporal patterns in dung beetle research. The supplemental file gives a complete list of sources used in the review analysis. Is there a reason you did not have a column listing the authors of papers in the supplemental file? I would like to see that included. The article is structured well and logically organized into subsections which align well between sections such as the structure of the discussion to address the aims outlined earlier in the paper.

Validity of the findings

Unfortunately I feel that the discussion and conclusion are lacking. The results and discussion section includes a discussion relating to some of the aims e.g. Q1, but no discussion of others, which simply state the results e.g. Q3. The discussion does not feel like a complete and well rounded argument. Furthermore, some areas of the discussion do not seem relevant to the results of the study, there is not a good sense of a well developed argument and the reader is left with a sense of 'so what?'. Some more specific points about the results and discussion section:
Line 206: you state 4 subgroups but only go on to explain 2 of them
Lines 212-227: this discussion is not directly relevant to the results and doesn't flow well as an argument, e.g. lines 226-227 seem like a random addition - explain how that is related to what you just discussed? Do not assume this is obvious to the reader, be clear and thorough.
Lines 299-316: Again it is not clear how this discussion directly links to your results.

The conclusion is weak and poorly worded. You need to identify a clear argument and tell a story with your results, ending in a well thought out conclusion. Unresolved questions are not identified in the conclusion. Also you mention 'three major discourses' in the conclusion for the first time - this is not explained or identified in the results and discussion. You should discuss earlier on in the paper how you identified these discourses, and also what a discourse is.

Additional comments

This has the potential to be a really neat paper, with some improvements to the discussion and conclusion I think it will be an interesting and valuable addition to the literature. I enjoyed reading it and found the methodology to be simple yet effective in producing clear and informative results. I am looking forward to reading the updated version - thank you to the authors for their hard work on this.

---

## Round 0.2 · Major Revisions

Dear authors, I ask you to further improve the manuscript in accordance with the recommendations of the reviewer.

·

Basic reporting

The authors took into account all the comments I made and made changes to the text of the manuscript. The article meets the PeerJ criteria and can be accepted.

Experimental design

No comments.

Validity of the findings

No comments.

Additional comments

No comments.

Reviewer 2 ·

Basic reporting

I checked the changes done to the manuscript. Since no major issues about the English text were detected in the first version of the paper, the text changes are acceptable.
References were added, and raw data are shared, thence the journal requirements about open access were addressed.Tables and figures are allright.
The review has a limitate interest (that is also declared by the authors), namely it is aimed at researchers working on dung beetles. Reviews about dung beetle literature were not recently published, thence the present review could be worthy of interest. As stated in my former comments, it could be useful to have an easily-available list of former literature on dung beetles, but the main issue is that a large part of the older papers are missing. Those are also the pieces of information harder to find for a beginner, thence it could be a major issue. The aim of the paper is given in general tems, but no details are furnished.

Experimental design

I'm not entirely sure that this paper fits within the scope of the journal. The focus is mostly within the 'history of research' field. The choice to examine the studies on dung beetles by an historical perspective is surely interesting, but the same results could be gained examining a different research field.
As a matter of fact it is quite common that along the years the interest of the researches shifted from some topics to others (irrespective the research field), and thence the analysis methods changed as well

although the data collection was thoroughly done, authors also declared that part of the information is missing, thence their study could be undermined by the lacking data. this could be an issue. In their reply authors stated that the missing data references are essentially grey literature (GL), but the missing information (books, papers not published in peer reviewed journals and so on) did not fall into this definition, since GL includes every information OUTSIDE of traditional publishing or distribution channels. the missing data are merely old references that are hard to collect.

Methods are described in details and could be replicated.

Validity of the findings

The findings are what expected (ithe evolution of a research topic along the years based on different perspectives and availability of new methods), thence the impact of the manuscript is average. The findings are not widely discussed. A more thorough discussion could had given more useful information to the readers.

Additional comments

I suggest to modify the focus of the paper into a historical review of the evolution of dung beetle research along the years, evaluating the different approaches, their merits and limits.

Cite this review as

---

## Round 0.3 · accepted · Accept

Dear authors, I am pleased to inform you that this article has been accepted for publication in our journal.

Reviewer 2 ·

Basic reporting

no comment

Experimental design

no comment

Validity of the findings

no comment

Additional comments

no comment

Cite this review as